# Effects of Hawthorn Fruit Supplementation on Facial Skin Phenotypes and Leukocyte Telomere Length Stratified by *TERT* Polymorphisms

**DOI:** 10.3390/nu17121983

**Published:** 2025-06-12

**Authors:** Minju Kim, Inkyung Baik

**Affiliations:** Department of Foods and Nutrition, College of Science and Technology, Kookmin University, Seoul 02707, Republic of Korea; km7985@kookmin.ac.kr

**Keywords:** *Crataegus*, skin, telomere, aging, clinical trial

## Abstract

Objectives: A randomized, double-blind, placebo-controlled intervention study aimed to evaluate whether hawthorn fruit (HF) supplementation can influence facial skin phenotypes and leukocyte telomere length (TL) and whether these effects differ by genetic polymorphisms related to TL. Participants/Methods: Among 41 male and female adults aged 25–75 years who participated in the study, 36 completed initial and follow-up examinations over 6 months. The HF supplementation group (*n* = 17) was instructed to take a powdered HF supplement (900 mg/day), while controls (*n* = 19) were to take a cornstarch placebo (900 mg/day). Facial skin phenotypes, including pigmentation, pores, hydration, wrinkles, and elasticity, were measured before and after the intervention, and changes in these phenotype scores were calculated. Sequencing of telomerase reverse transcriptase (*TERT*) polymorphisms, such as rs7705526 (C>A) and rs2853669 (A>G), was conducted. Results: The HF supplementation group exhibited significantly improved hydration scores compared to the control group; the mean changes (follow-up measure—baseline measure) [standard deviation] in hydration scores over 6 months were 1.71 [8.18] and −3.00 [8.42] for the supplementation group and control group, respectively (*p* < 0.05) (Cohen’s *d* = 0.57). However, changes in other phenotypes and leukocyte TL were similar between groups. The genotype-specific analysis revealed that the improvement of hydration state was most noticeable among carriers with the CC genotype of rs7705526 (*p* < 0.05) (Cohen’s *d* = 1.50) and that the HF supplementation group exhibited reduced wrinkle scores while the control group showed increased scores among carriers of the AA genotype of rs2853669 (*p* < 0.05) (Cohen’s *d* = 1.40). In correlation analysis for all participants, hydration scores were positively correlated with leukocyte TL (Spearman correlation coefficient: 0.36; *p* < 0.05). Conclusions: These findings suggest that HF consumption may have potential anti-skin-aging effects. Future studies may need to elucidate the biological mechanisms underlying these effects.

## 1. Introduction

Skin changes related to aging include wrinkles, dryness, pigmentation, and loss of elasticity. These phenotypes of skin aging are determined by multiple risk factors, such as age, gender, ethnicity, genetic factors, ultraviolet (UV) radiation from sunlight, exposures to air pollution or toxic substances, smoking, and inadequate diet [1]. Particularly, modifiable factors including diet or nutraceuticals are of interest to peoples who want to improve their skin. A number of clinical trials have attempted to evaluate the effects of nutraceutical components, such as collagen peptide, polysaccharides, and botanical extracts on skin improvement [2].

Hawthorn is a deciduous tree, belonging to the *Crataegus* genus in the Rosaceae family, with a well-established history as a medicinal herb. Hawthorn fruit (HF) contains various bioactive compounds, such as polyphenolic compounds, fatty acids, and organic acids [3]. In an experimental study, in which mouse models with skin damage were treated with a high dose of HF extract, their skin moisture increased by 34% [4]. In a randomized, double-blind clinical trial involving middle-aged or older women with photodamage, a mixture of ginseng and HF extracts applied on skin significantly improved skin roughness and smoothness depth [5]. However, whether oral HF supplementation can improve skin phenotypes remains unclear.

In earlier experimental studies, rats orally administered HF extract showed longer leukocyte telomere length (TL), regarded as a marker of biological age, compared with untreated controls, along with increased serum activity of telomerase [6,7], which is expected to maintain TL by adding telomere nucleotide repeats at the ends of chromosomes [8]. It was suggested that leukocyte TL is likely to be a surrogate for skin TL, based on data that average TLs in leukocytes and skin cells have a linear correlation and both are inversely correlated with age [9,10]. However, data on the association between skin phenotypes and leukocyte TL are limited. Furthermore, whether HF supplementation can improve skin phenotypes and whether this effect can be modified by genetic polymorphisms related to TL remain unclear; notably, such hypotheses have not yet been evaluated in human studies.

The current intervention study aimed to evaluate the bioactive effects of HF supplementation on facial skin phenotypes and leukocyte TL in healthy adults. Furthermore, the study assessed whether these effects differ by the polymorphisms of the telomerase reverse transcriptase gene (*TERT*), which plays a role in telomere biology [11].

## 2. Participants and Methods

### 2.1. Study Design and Participants

This study was designed as a randomized, double-blind, placebo-controlled, two-arm parallel-group superiority trial and conducted in Kookmin University (Seoul, Republic of Korea). Sixty-three participants were recruited via social media and posters from June to July 2022, of which 48 were invited to visit the study site in response to a telephone interview using a screening questionnaire with the inclusion and exclusion criteria. On the basis of an earlier study [5], we performed a power analysis using the G*Power program (version 3.1.9.2; HHU, Düsseldorf Universität, Düsseldorf, Germany). Thus, a sample of 36 subjects (18 for each group) would provide an 80% power at a two-sided significance level of 5% for the non-parametric test. Healthy male and female adults aged 25–75 years were eligible. Those who had chronic diseases (cardiovascular disease, cancer, hepatic disease, and kidney failure), allergic symptoms after using herb supplements, or an anemia diagnosis in the last 3 months, who were regularly taking medications to treat diseases, or who were pregnant or lactating women were excluded. Forty-one participants were enrolled after signing an informed consent form prior to participating in the study, which was conducted according to the guidelines of the Declaration of Helsinki and was approved by the Institutional Review Board of Kookmin University (KMU-202204-HR-305, 4 May 2022). The protocol of this study, which includes a plan to halt the study if side effects are considered to be greater than minimal risk, was registered on 10 June 2025 in the Clinical Research Information Service (KCT0010611). These participants were sequentially numbered and randomly allocated using a stratified randomization according to age and sex to either a control group (*n* = 21) or an HF supplementation group (*n* = 20) with allocation concealment. The researcher who conducted the random allocation of participants was not allowed to be involved in interviewing and assessing outcomes. The control group received a placebo supplement (900 mg of cornstarch), while the HF supplementation group received 900 mg of HF supplement (Hawthorn Berry Extract Powder, BulkSupplements.com, Henderson, NV, USA: 100% *C. pinnatifida* fruit extract). All participants were instructed to take supplements every day, 2 h before bedtime, for 6 months. To adhere to the supplementation regimen, each participant received a 30-pocket-calendar containing the placebo or HF supplements and a monthly phone call from researchers. Researchers who were blinded for the group allocation conducted baseline and follow-up assessments with a 6-month interval. Although no side effects were reported during this period, five participants did not participate in the follow-up assessment due to time constraints in their personal schedule. Accordingly, a total of 36 participants (*n* = 19 in the control group and *n* = 17 in the HF supplementation group) completed the baseline and follow-up assessments (Figure 1).

### 2.2. Questionnaire Data Collection

Questionnaires acquired information on participant smoking status, physical activity, and side effects of the supplement. For physical activity, the total metabolic equivalent of task-hours (MET-h) per day were calculated by multiplying MET values by hours spent in sleep and five categories of activity intensity (1.0 for sleep or sedentary, 1.5 for very light activity, 2.4 for light activity, 5.0 for moderate activity, and 7.5 for vigorous activity).

### 2.3. Anthropometric Measurement

Anthropometric measurements were taken in light clothing without shoes and personal belongings. Body weight (kg) and height (cm) were measured to the nearest 0.1 kg and 0.1 cm, respectively, to calculate body mass index (BMI, kg/m^2^). Waist circumference (cm) was measured using a flexible measuring tape and rounded to the nearest 0.1 cm.

### 2.4. Assessment of Facial Skin Phenotypes

Facial skin phenotypes, such as pigmentation, pores, hydration, wrinkles, and elasticity, were measured using a skin analyzer (Bomtech, Seoul, Republic of Korea); pigmentation in the skin was measured 1 cm below the corner of the left eye, pores on the cheek in contact with the left wings of the nose, and hydration state, wrinkles, and elasticity on the middle of the forehead. The levels of each phenotype were automatically calculated into scores.

### 2.5. Assay of Leukocyte Telomere Length

Whole blood samples were collected from participants who had fasted for at least 8 h for assays of leukocyte TL. Genomic DNA was extracted from leukocytes using the DNeasy Blood and Tissue Kit (Qiagen, Hilden, Germany) and quantified using a NanoDrop 1000 spectrophotometer (Thermo Fisher Scientific, Wilmington, DE, USA). Relative leukocyte TL was assayed in each DNA sample based on the prior method [12,13]. In essence, the ratio of the telomere repeat copy number to the single-copy gene copy number (*36B4* encoding acidic ribosomal phosphoprotein) was determined using the CFX96 Touch RT-PCR detection system (Bio-Rad, Hercules, CA, USA). The final concentrations of PCR reagents were 1× SYBR Green SuperMix (Bio-Rad), 50 ng DNA, 0.2 μM telomere primers, and 0.3 μM *36B4* primers.

### 2.6. Collection of DNA Samples for Genotyping

Oral epithelial cell samples were collected from participants using a buccal swab collection kit. Participants rinsed their mouths with water 30 min before collection. DNA collectors were rubbed five times on both sides of the inner cheeks in a direction from the back to the front part of the mouth. Labeled collection kits were packaged and delivered to a commercial laboratory (Theragen Bio, Seongnam, Republic of Korea) for sequencing of *TERT* polymorphisms: rs7705526 (C>A) and rs2853669 (A>G). These polymorphisms were reportedly associated with TL in earlier genome-wide and candidate gene association studies [14,15].

### 2.7. Statistical Analysis

Descriptive statistics were presented as mean ± standard deviation or proportion. Comparisons between control and HF supplementation groups were conducted by performing a chi-square or Fisher’s exact test for categorical variables and Student’s *t*-test or the non-parametric Wilcoxon rank-sum test for continuous variables. Differences between baseline and follow-up data were calculated and compared between control and HF supplementation groups using the Wilcoxon signed rank test. In addition, Cohen’s *d* was calculated as a measure of effect size; Cohen’s *d* values greater than 0.8 are considered as a large effect. Statistical significance was set at 0.05. The SAS v.9.4 software (SAS Institute Inc., Cary, NC, USA) was used to analyze the data.

## 3. Results

### 3.1. Comparison of Baseline Characteristics Between Control and HF Supplementation Groups

The baseline characteristics between control and HF supplementation groups were compared in the intention-to-treat analysis for 41 participants as well as in the per-protocol analysis for 36 participants (Table 1). No significant differences between groups were observed in the intention-to-treat analysis.

### 3.2. Comparison of Baseline and Follow-Up Data Between Control and HF Supplementation Groups

In Table 2, baseline and follow-up data of facial skin phenotypes and leukocyte TL, as well as absolute changes, which were calculated by subtracting baseline data from follow-up data, were compared between the control and HF supplementation groups. While the control group showed higher facial skin hydration scores than the HF supplementation group at baseline (*p* < 0.05), no significance was observed in the follow-up hydration scores between groups; the mean scores decreased in the control group, but increased in the HF supplementation group. Accordingly, changes in hydration scores differed significantly between groups (*p* < 0.05) (Cohen’s *d* = 0.57). In addition, skin wrinkle scores did not differ between groups at baseline. However, they differed significantly at the end of follow-up (*p* < 0.05) because wrinkle scores increased in the control group and decreased in the HF supplementation group, although their changes were similar. Changes in other phenotypes and leukocyte TL were similar between groups.

### 3.3. Correlations Between Relative Leukocyte Telomere Length Changes and Facial Skin Hydration Score Changes

The mean changes in skin hydration scores decreased, while the relative leukocyte TL changes increased among all participants. Figure 2 presents scatter plots between changes in the relative leukocyte TL and changes in facial skin hydration scores with a trend line. A significant and positive correlation was found between these change values; the Spearman correlation coefficient was 0.36 (*p* < 0.05).

### 3.4. Comparison of the TERT Polymorphism Genotype Frequencies Between Control and HF Supplementation Groups

Table 3 demonstrates the genotype frequencies of the rs7705526 and rs2853669 *TERT* polymorphisms between the control and HF supplementation groups. In the genotype frequencies of rs7705526, carriers with CC comprised 42% of all, 53% of controls, and 29% of participants in the supplementation group. In the genotype frequencies of rs2853669, carriers with AA comprised 36% of all, 37% of controls, and 35% of participants in the supplementation group. No significant genotype distributions of these polymorphisms were observed between the control and HF supplementation groups.

Table 4 presents the changes in facial skin phenotypes and relative leukocyte TL compared between the control and HF supplementation groups according to the polymorphic genotypes. Among carriers with the CC genotype of rs7705526, the HF supplementation group showed significantly increased hydration scores compared with the control group (*p* < 0.05) (Cohen’s *d* = 1.50). In addition, the HF supplementation group exhibited lower wrinkle scores than the control group among carriers of the AA genotype of rs2853669 (*p* < 0.05) (Cohen’s *d* = 1.40). Among carriers with the CA and AA genotypes of rs7705526 or those with the AG and GG genotypes of rs2853669, no differences in other skin phenotypes were observed between the control and HF supplementation groups. In addition, changes in the relative leukocyte TL did not differ between groups across the polymorphic genotypes.

## 4. Discussion

A randomized, double-blind, placebo-controlled study evaluated whether HF supplementation influences facial skin phenotypes and leukocyte TL and whether these effects are modified by *TERT* polymorphisms known to be related to TL maintenance. The HF supplementation group exhibited significantly increased skin hydration scores compared with the control group. Particularly, this supplemental benefit was revealed exclusively among carriers with the major homozygous allele of rs7705526. In addition, the supplementation lowered wrinkles among carriers with the major homozygous allele of rs2853669. Although leukocyte TL did not differ between groups, leukocyte TL changes were positively correlated with changes in skin hydration scores among all participants.

Hawthorn (*Crataegus* spp.) is distributed across Europe, North Africa, North America, and Asia [16]. *C. pinnatifida* is a dominant cultivated species in Asia, and its traditional use due to medicinal properties, such as treatment of indigestion, hernia, and blood stasis, is widely recognized [3]. It was reported that HF contains high polyphenolic compounds including more than 50 flavonoids; its total phenolic content ranged from 2 to 249 mg/g and total flavonoid content was estimated to be up to 147 mg/g [17]. Recently, accumulating data from clinical trials indicate the beneficial effects of hawthorn consumption on cardiovascular diseases [18]. However, data on the effects of hawthorn consumption on skin are limited.

Two studies conducted in vitro and in vivo investigated using polyphenol extract from HF [4,19]. In the in vitro experiments, the extract increased cell viability in UV-irradiated HaCaT cells (immortalized human keratinocytes) and human dermal fibroblasts, whereas it decreased the production of reactive oxidative species and the expression of matrix metalloproteinase, which is known to be involved in collagen degradation and wrinkle formation leading to skin aging. In addition, in the in vivo experiments, applying the extract to the UV-irradiated skin of mouse models revealed that the extract decreased dermal damage [4,19].

Earlier clinical trials evaluated the effects of the consumption of fruits or fruit extract, including red orange, green mandarin, lingonberry, amla, pomegranate, and avocado, on skin aging in adults [20]. There is a single clinical trial investigating the effects of HF extract on skin [5]. In a randomized, double-blind, and placebo-controlled study involving 20 adult participants with photodamage, application of a topical cream containing a mixed extract of ginseng and HF to their eyelid skin was found to improve skin roughness associated with moisture loss compared with the control group. To the best of our knowledge, there are no human data on the exclusive effects of hawthorn on skin phenotypes, especially on the effects of HF consumption on skin health. Furthermore, our study attempted to analyze the impact of genetic polymorphisms related to telomere biology on the responses to skin phenotypes by HF consumption.

Telomeres are composed of repetitive DNA sequences (TTAGGG) and protect chromosomal ends, but TL progressively shortens with each cell division because of the end-replication problem [21]. A genome-wide association study identified 20 genomic loci associated with leukocyte TL, including *TERT* [15]. Furthermore, the minor alleles of the *TERT* rs7705526 and rs2853669 are reportedly associated with shorter leukocyte TL in Koreans [14]. Because TERT is a subunit of the enzyme telomerase, the upregulation of *TERT* expression leads to increased telomerase activity and telomere maintenance, including TL extension [22]. Although telomerase activity is typically deficient in most normal somatic cells, it was reportedly detected in the epidermis; particularly, it was detected in normal keratinocytes, which are the most abundant type of epidermal cells and play a role as a barrier preventing water loss and protecting against external damages [23]. It was reported that the rates of telomere shortening are similar in adult tissues, including leukocytes, skin, skeletal muscle, and subcutaneous fat [24] and that TLs in leukocytes and skin cells have a linear correlation regardless of age [9]. Accordingly, we assessed the relative leukocyte TL as a surrogate biomarker of skin TL because the direct assessment of skin TL or epidermal telomerase activity is not feasible in a human study.

Our study revealed a significant improvement of skin hydration in the HF supplementation group compared to the control group, with a positive correlation between skin hydration score changes and leukocyte TL changes. In genotype-specific analysis, the beneficial effects of HF supplementation on skin phenotypes, such as improved skin hydration and reduced wrinkles, were significant among carriers with the major alleles of rs7705526 and rs2853669 (the CC genotype of rs7705526 and the AA genotype of rs2853669), but not among those with the mutant alleles. Based on these results, because the Cohen’s d values indicate a large effect, the supplementation was likely to have a meaningful impact on skin hydration and wrinkle scores. However, changes in leukocyte TL did not differ between the groups in overall and genotype-specific analyses. The sample size, the intervention period, and the supplement doses of our study might be insufficient to achieve significant TL changes in the HF supplementation group. Nevertheless, because the genotype-specific beneficial effects of HF supplementation on skin phenotypes were significant, we speculate that carriers with the major alleles of rs7705526 and rs2853669 might have upregulation of *TERT* expression via HF supplementation, leading to increased TERT, which is known to play a role in the antioxidant defense mechanisms besides its role in TL extension [25] in the epidermis. It has been suggested that TERT and telomerase can be modulated by external damages related to skin aging as well as by diet or nutraceutical compounds, especially those in herbs such as polyphenol compounds, flavonoids, and organic acids with antioxidant properties [26].

The limitations of this study include the small sample size, the assessment of leukocyte TL, which was used as a surrogate for skin TL because of ethical concerns, and the limited generalizability of the findings. Despite these limitations, the strength of this study is the randomized, double-blind, placebo-controlled design and exhibition of intervention effects stratified by genetic polymorphisms, which may be informative for formulating personalized anti-aging functional foods in the future. Further studies are needed to explore the biological mechanisms underlying the effects of HF supplementation on skin phenotypes.

## 5. Conclusions

This randomized, double-blind, placebo-controlled study evaluated the effects of HF supplementation on facial skin phenotypes. At the end of the 6-month follow-up period, the HF supplementation group showed an increased hydration state and reduced wrinkle formation, whereas the control group had reduced hydration and increased wrinkles. In particular, such beneficial effects of HF supplementation were significant among carriers with particular genetic polymorphisms related to telomere biology. These results suggest that HF supplementation may help improve skin aging and further imply that personalized functional foods based on the information of genetic profiling may optimize benefits for health and wellness. At this time, however, more data regarding the genotype-specific effects are warranted to apply a recommendation of personalized functional foods.

## Figures and Tables

**Figure 1 nutrients-17-01983-f001:**
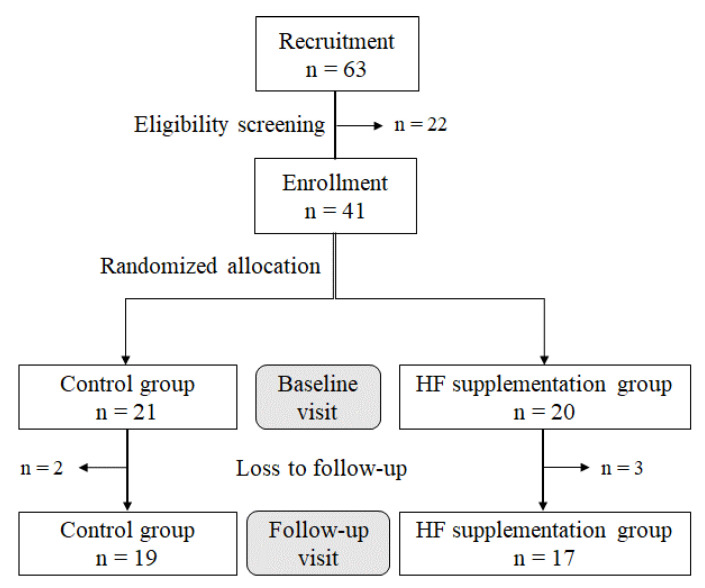
Flow chart showing the participants in the study. Abbreviation: HF, hawthorn fruit.

**Figure 2 nutrients-17-01983-f002:**
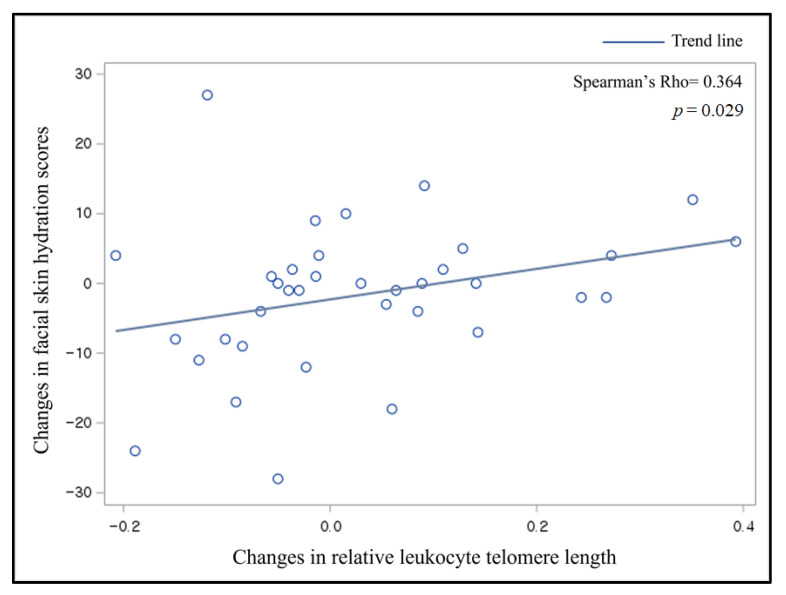
Scatter plots and a trend line for the relation between changes in leukocyte telomere length and changes in facial skin hydration scores.

**Table 1 nutrients-17-01983-t001:** Comparison of baseline characteristics between the control and HF supplementation groups according to the analytical method.

Variables		Per–Protocol Analysis			Intention–to–Treat Analysis		
		Control	HF Supplementation	*p*	Control	HF Supplementation	*p*
Number of participants		19	17		21	20	
Age, years		40.3 ± 13.9	46.8 ± 15.6	0.228	40.4 ± 14.2	44.8 ± 15.3	0.354
Sex, *n* (%)	Male	8 (42.1)	7 (41.2)	0.955	9 (42.9)	7 (35.0)	0.606
	Female	11 (57.9)	10 (58.8)		12 (57.1)	13 (65.0)	
Weight, kg		66.7 ± 14.6	62.4 ± 11.0	0.447	66.0 ± 15.5	60.7 ± 11.2	0.215
Height, cm		167.2 ± 8.5	164.4 ± 8.6	0.428	167.0 ± 8.6	163.8 ± 8.3	0.239
Body mass index, kg/m^2^		23.6 ± 4.0	23.0 ± 3.5	0.751	23.4 ± 4.2	22.6 ± 3.5	0.470
Waist measurement, cm		83.3 ± 10.6	80.5 ± 9.9	0.466	82.4 ± 10.7	79.1 ± 9.8	0.309
Smoking status, *n* (%)	Nonsmoker	13 (68.4)	14 (82.4)	0.451	15 (71.4)	17 (85.0)	0.454
	Smoker	6 (31.6)	3 (17.6)		6 (28.6)	3 (15.0)	
Total physical activity, MET–h		42.8 ± 8.3	43.4 ± 7.5	0.668	43.5 ± 8.6	44.2 ± 8.0	0.789

Abbreviations: HF, hawthorn fruit; MET, metabolic equivalent of task. Values are presented as mean ± standard deviation or number (%). Comparisons were performed using the chi-square or Fisher’s exact test for categorical variables and Wilcoxon rank-sum test for continuous variables.

**Table 2 nutrients-17-01983-t002:** Comparison of baseline and follow-up data between the control and HF supplementation groups.

Variables	Baseline Data		Follow-Up Data		Change Data *	
	Control	HF Supplementation	Control	HF Supplementation	Control	HF Supplementation
Skin phenotype scores						
Pigmentation	6.47 ± 3.79	6.35 ± 3.46	5.11 ± 3.16	5.65 ± 3.32	−1.37 ± 4.45	−0.71 ± 4.27
Pores	6.37 ± 4.07	5.35 ± 4.50	6.58 ± 3.88	4.35 ± 4.31	0.21 ± 3.88	−1.00 ± 5.15
Hydration	34.53 ± 7.04	30.35 ± 7.01 ^†^	31.53 ± 5.19	32.06 ± 3.91	−3.00 ± 8.42	1.71 ± 8.18 ^†^
Wrinkles	3.11 ± 2.21	2.53 ± 1.74	4.16 ± 1.71	2.89 ± 2.03 ^†^	1.05 ± 2.66	0.35 ± 2.12
Elasticity	20.74 ± 12.18	24.24 ± 9.59	21.95 ± 14.00	25.59 ± 14.18	1.21 ± 15.36	1.35 ± 18.72
Relative leukocyte TL	0.71 ± 0.33	0.69 ± 0.24	0.74 ± 0.32	0.72 ± 0.34	0.03 ± 0.13	0.03 ± 0.16

Abbreviation: HF, hawthorn fruit; TL, telomere length. Values are presented as mean ± standard deviation. Comparisons were performed using the Wilcoxon rank-sum test or Wilcoxon signed rank test. * Change data were calculated by subtracting baseline data from follow-up data. ^†^  *p* < 0.05 when compared with the control group.

**Table 3 nutrients-17-01983-t003:** Comparison of genotype frequencies of *TERT* polymorphisms between the control and HF supplementation groups.

Variables	All (*n* = 36)	Control (*n* = 19)	HF Supplementation (*n* = 17)	*p* Between Groups
Genotypes of rs7705526, *n* (%)				0.393
CC	15 (41.67)	10 (52.63)	5 (29.41)	
CA	16 (44.44)	7 (36.84)	9 (52.94)	
AA	5 (13.89)	2 (10.53)	3 (17.65)	
Genotypes of rs2853669, *n* (%)				0.636
AA	13 (36.11)	7 (36.84)	6 (35.29)	
AG	21 (58.33)	10 (52.63)	11 (64.71)	
GG	2 (5.56)	2 (10.53)	0 (0.00)	

Abbreviations: *TERT*, telomerase reverse transcriptase gene; HF, hawthorn fruit. Values are presented as number (%). Comparisons were performed using Fisher’s exact test.

**Table 4 nutrients-17-01983-t004:** Comparison of changes in facial skin phenotype scores and relative leukocyte telomere length stratified by the *TERT* polymorphic genotypes between the control and HF supplementation groups.

*TERT* Polymorphisms	Groups	Changes * in Facial Skin Phenotype Scores					Changes in LTL
		Pigmentation	Pores	Hydration	Wrinkles	Elasticity	
Genotypes of rs7705526							
CC	Control	−2.60 ± 3.57	−0.70 ± 4.79	−4.30 ± 9.44	1.30 ± 2.36	3.30 ± 19.18	−0.005 ± 0.12
	HF supplementation	−5.20 ± 2.68	−4.20 ± 4.60	5.80 ± 1.10	−0.60 ± 2.70	9.00 ± 24.84	0.06 ± 0.22
	*p* between groups	0.094	0.174	0.016	0.322	0.759	0.501
CA and AA	Control	0.01 ± 5.12	1.22 ± 3.07	−1.56 ± 7.40	0.78 ± 3.07	−1.11 ± 10.26	0.06 ± 0.15
	HF supplementation	1.17 ± 3.30	0.33 ± 4.92	0.01 ± 9.28	0.75 ± 1.82	−1.83 ± 15.72	0.02 ± 0.14
	*p* between groups	0.775	0.504	0.498	0.773	0.972	0.374
Genotypes of rs2853669							
AA	Control	−0.43 ± 4.12	0.01 ± 5.20	−3.14 ± 5.30	2.86 ± 2.12	1.43 ± 19.56	0.04 ± 0.12
	HF supplementation	−2.33 ± 4.23	−0.50 ± 5.86	3.00 ± 4.20	−0.50 ± 2.66	−0.50 ± 24.42	−0.001 ± 0.13
	*p* between groups	0.428	0.763	0.050	0.030	0.721	0.721
AG and GG	Control	−1.92 ± 4.72	0.33 ± 3.14	−2.92 ± 10.03	0.01 ± 2.41	1.08 ± 13.31	0.02 ± 0.14
	HF supplementation	0.18 ± 4.21	−1.27 ± 5.00	1.00 ± 9.84	0.82 ± 1.72	2.36 ± 16.10	0.05 ± 0.18
	*p* between groups	0.422	0.290	0.338	0.198	0.782	0.976

Abbreviations: *TERT*, telomerase reverse transcriptase gene; HF, hawthorn fruit; LTL, leukocyte telomere length. Values are presented as mean ± standard deviation. Comparisons were performed using the Wilcoxon rank-sum test. * Changes were calculated by subtracting baseline data from follow-up data.

## Data Availability

The de-identified participant data (including data dictionary), statistical code, and any other materials (including study protocol and intervention manual) are available on reasonable request from the corresponding author due to ethical reasons.

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
