# Peer review of "Effects of Hawthorn Fruit Supplementation on Facial Skin Phenotypes and Leukocyte Telomere Length Stratified by TERT Polymorphisms"

_nutrients, 2025, doi:10.3390/nu17121983_

Round 1
Reviewer 1 Report
Comments and Suggestions for Authors
This manuscript presents the results of a randomized, double-blind, placebo-controlled study evaluating the impact of hawthorn fruit (HF) supplementation on facial skin phenotypes and leukocyte telomere length (TL), while accounting for genetic variability in telomerase reverse transcriptase (TERT) polymorphisms. The study is innovative in its integration of dermatological assessment, telomere biology, and personalized genetic analysis, offering a unique perspective on the potential anti-aging effects of plant-derived supplementation.
The manuscript follows a classical scientific structure, with well-organized content and clearly presented tables and figures. The topic is timely and highly relevant, particularly in the context of personalized nutrition and aging research. The study is well designed, incorporating appropriate randomization, double-blinding, placebo control, and a 6-month intervention period. The results are clearly reported, and visual data representations are easy to interpret.
However, the composition of the HF supplement is not sufficiently described. Providing a detailed phytochemical profile (e.g., total polyphenol or flavonoid content) would enhance reproducibility and aid in the interpretation of findings.
Several areas should be improved:
The abstract requires revision for clarity and conciseness. Some sentences are unnecessarily complex, and the presentation of findings should be better structured—starting with overall comparisons and then genotype-specific results. Including effect sizes and confidence intervals would also improve transparency.
The conclusion section mostly restates the results and should be rewritten to provide a more interpretive summary. It should better emphasize the relevance of genotype-specific responses and suggest possible clinical implications and directions for further research, while briefly acknowledging the study’s limitations.
The discussion includes background information on hawthorn fruit that would be more appropriate in the introduction. Instead, this section should focus on a more in-depth interpretation of the findings in the context of the current literature. Potential biological mechanisms underlying the observed effects—particularly those involving the interaction between HF supplementation and TERT-related telomere biology—should be more thoroughly explored. Overall, the interpretation lacks depth and critical analysis and would benefit from further development.
Author Response
Manuscript ID: nutrients-3674791
Title: Effects of Hawthorn Fruit Supplementation on Facial Skin Phenotypes and Leukocyte Telomere Length Stratified by TERT Polymorphisms
We are most grateful with the reviewers’ comments on our manuscript. We have revised the manuscript in accordance with the comments. We list here our response to each comment.
Response to Reviewer 1’s comments
- The composition of the HF supplement is not sufficiently described. Providing a detailed phytochemical profile (e.g., total polyphenol or flavonoid content) would enhance reproducibility and aid in the interpretation of findings.
Response: We appreciate the reviewer's comments. We have now added more information on this specific supplement (page 2, revised) as well as on the composition of the HF previously reported (page 9, revised).
- The abstract requires revision for clarity and conciseness. Some sentences are unnecessarily complex, and the presentation of findings should be better structured—starting with overall comparisons and then genotype-specific results. Including effect sizes and confidence intervals would also improve transparency.
Response: According to the reviewer’s suggestion, we have now revised the abstract (page 1, revised). In addition, we have now added Cohen’s d as a measure of effect size (pages 1, 4, and 7, revised).
- The conclusion section mostly restates the results and should be rewritten to provide a more interpretive summary. It should better emphasize the relevance of genotype-specific responses and suggest possible clinical implications and directions for further research, while briefly acknowledging the study’s limitations.
Response: According to the reviewer’s suggestion, we have now revised the conclusion (page 10, revised).
- The discussion includes background information on hawthorn fruit that would be more appropriate in the introduction. Instead, this section should focus on a more in-depth interpretation of the findings in the context of the current literature. Potential biological mechanisms underlying the observed effects—particularly those involving the interaction between HF supplementation and TERT-related telomere biology—should be more thoroughly explored. Overall, the interpretation lacks depth and critical analysis and would benefit from further development.
Response: According to the reviewer’s suggestion, we have now added a more interpretation and explanation of biological mechanisms (pages 9-10, revised).
Reviewer 2 Report
Comments and Suggestions for Authors
This randomized, double-blind, placebo-controlled study explores whether hawthorn fruit (HF) supplementation has an impact on facial skin phenotypes and leukocyte telomere length (TL), and whether these effects are influenced by TERT gene polymorphisms. The authors report improvements in skin hydration and reductions in wrinkles among specific genotype carriers, along with a positive correlation between TL and skin hydration.
The manuscript presents several strengths. It effectively integrates dermatological outcomes with genetic data—specifically TERT polymorphisms—within the context of a nutritional intervention. The use of a randomized, double-blind, placebo-controlled design adds methodological rigor, and the inclusion of diverse outcomes such as skin phenotype measurements, telomere length, and genotyping enhances the biological plausibility and supports mechanistic interpretations. The study aligns well with the scope of Nutrients, particularly in the areas of aging, nutricosmetics, and personalized nutrition.
However, the manuscript also presents several weaknesses that should be addressed. The relatively small final sample size (n = 36) limits both the statistical power and the generalizability of the findings. Subgroup analyses based on genotype are especially underpowered, which raises concerns about the robustness of those results. While some changes in hydration and wrinkle scores reached statistical significance, the actual magnitude of these effects appears modest and may not translate into clinically meaningful improvements. Furthermore, changes in TL were not significant between the intervention and control groups, and skin TL was not measured directly, which weakens the mechanistic conclusions related to telomere biology. Baseline differences in hydration scores between groups also complicate the interpretation of post-intervention results. In addition, there is insufficient information on participant adherence to the supplementation regimen beyond the dropout rate; the inclusion of measures such as pill counts or biomarker data would help validate compliance. The potential influence of expectancy or placebo effects on subjective skin outcomes is also not discussed.
In my opinion, the manuscript would benefit from several improvements. A power analysis should be added to the methods section, and the limitations related to sample size should be more thoroughly discussed. The authors should also consider using ANCOVA or other statistical techniques that adjust for baseline differences when comparing follow-up scores. It would be important to provide more detailed information on how compliance with supplementation was monitored. Furthermore, the authors should clarify whether the observed changes in hydration and wrinkle scores are likely to be perceptible or meaningful from a clinical or cosmetic standpoint. While the positive correlation between TL and hydration is an interesting finding, any implication of causality should be moderated given the absence of significant group differences in TL. Finally, the inclusion of additional visual representations, such as bar graphs with error bars to illustrate genotype-specific changes, would enhance the clarity and impact of the results.
Overall, the study is well-designed and fits within the journal’s scope. Its focus on genetic stratification in a nutritional context is both timely and innovative. Nevertheless, the limitations related to sample size and baseline imbalances must be more carefully addressed through clearer analysis and expanded discussion to strengthen the manuscript’s validity and interpretability.
Author Response
Manuscript ID: nutrients-3674791
Title: Effects of Hawthorn Fruit Supplementation on Facial Skin Phenotypes and Leukocyte Telomere Length Stratified by TERT Polymorphisms
We are most grateful with the reviewers’ comments on our manuscript. We have revised the manuscript in accordance with the comments. We list here our response to each comment.
Response to Reviewer 2’s comments
- The manuscript also presents several weaknesses that should be addressed. The relatively small final sample size (n = 36) limits both the statistical power and the generalizability of the findings. Subgroup analyses based on genotype are especially underpowered, which raises concerns about the robustness of those results. While some changes in hydration and wrinkle scores reached statistical significance, the actual magnitude of these effects appears modest and may not translate into clinically meaningful improvements. Furthermore, changes in TL were not significant between the intervention and control groups, and skin TL was not measured directly, which weakens the mechanistic conclusions related to telomere biology.
Response: We appreciate the reviewer's comments. We understand the reviewer’s concern about the robustness of the genotype-specific results due to a small sample size. To ease this concern, we have now added the information of a measure of effect size (Cohen’s d) (pages 1, 4, and 7, revised). The Cohen’s d values are considered a large effect, indicating that the supplementation was likely to have a meaningful impact on skin hydration and wrinkle scores (pages 4, 7, revised). The study weaknesses that the reviewer mentioned, including a small sample size, limited generalizability of the findings, and not measuring skin TL directly, have been now discussed more (pages 9-10, revised).
- Baseline differences in hydration scores between groups also complicate the interpretation of post-intervention results. In addition, there is insufficient information on participant adherence to the supplementation regimen beyond the dropout rate; the inclusion of measures such as pill counts or biomarker data would help validate compliance. The potential influence of expectancy or placebo effects on subjective skin outcomes is also not discussed.
Response: According to the reviewer’s comments, we have now added more information on regimens to adhere the supplementation or double-blind regimens to minimize the potential influence of expectancy or placebo effects on subjective skin outcomes (pages 2-3. revised).
- In my opinion, the manuscript would benefit from several improvements. A power analysis should be added to the methods section, and the limitations related to sample size should be more thoroughly discussed. The authors should also consider using ANCOVA or other statistical techniques that adjust for baseline differences when comparing follow-up scores.
Response: We appreciate the reviewer's valuable suggestions. We have now added information of power calculation (page 2, revised) and discussed the limitation of a small sample size (pages 10, revised). In addition, we have now revised Table 2, which presents the comparison of changes between follow-up and baseline measures because of baseline differences (Table 2, revised).
- It would be important to provide more detailed information on how compliance with supplementation was monitored. Furthermore, the authors should clarify whether the observed changes in hydration and wrinkle scores are likely to be perceptible or meaningful from a clinical or cosmetic standpoint. While the positive correlation between TL and hydration is an interesting finding, any implication of causality should be moderated given the absence of significant group differences in TL. Finally, the inclusion of additional visual representations, such as bar graphs with error bars to illustrate genotype-specific changes, would enhance the clarity and impact of the results. Overall, the study is well-designed and fits within the journal’s scope. Its focus on genetic stratification in a nutritional context is both timely and innovative. Nevertheless, the limitations related to sample size and baseline imbalances must be more carefully addressed through clearer analysis and expanded discussion to strengthen the manuscript’s validity and interpretability.
Response: We have now added information on how compliance with supplementation was monitored (pages 2-3, revised). To clarify the impact of the supplementation on skin hydration and wrinkle scores, we have now added the information of a measure of effect size (Cohen’s d) (pages 1, 4, and 7, revised) and discussed the results (page 10, revised). But, we are sorry not to include additional visual representation for the genotype-specific changes because of duplication of results.